# Warming in Cold Seasons Increases the Abundance of Ground-Dwelling Collembola in Permafrost Wetlands

**DOI:** 10.3390/insects14010033

**Published:** 2022-12-30

**Authors:** Shaoqing Zhang, Zhijing Xie, Yongjing Dou, Xin Sun, Liang Chang, Donghui Wu

**Affiliations:** 1Key Laboratory of Wetland Ecology and Environment, Northeast Institute of Geography and Agroecology, Chinese Academy of Sciences, Changchun 130012, China; 2University of Chinese Academy of Sciences, Beijing 101408, China; 3Key Laboratory of Vegetation Ecology, Ministry of Education, Northeast Normal University, Changchun 130024, China; 4Department of Geography, Taiyuan Normal University, Taiyuan 030621, China; 5Key Laboratory of Urban Environment and Health, Institute of Urban Environment, Chinese Academy of Sciences, Xiamen 361024, China; 6Jilin Provincial Key Laboratory of Animal Resource Conservation and Utilization, Northeast Normal University, Changchun 130117, China; 7State Environmental Protection Key Laboratory of Wetland Ecology and Vegetation Restoration, Northeast Normal University, Changchun 130024, China

**Keywords:** soil animals, community assembly, permafrost wetland, climate change, open top chamber, high latitude

## Abstract

**Simple Summary:**

Global warming could affect Collembola and related decomposition processes within soil ecosystems in permafrost wetlands. Open top chambers (OTCs) were used to simulate climate warming in a cold temperate monsoon climate zone in the Great Hing’an Mountains of Northeast China. Collembola were captured using an aspirator after five years of simulated warming. We found that warming treatment increased the species richness and abundance of Collembola in most of the different seasons, except in May. Species composition differed significantly in the control and warming treatment in May and September. The Collembola species composition in permafrost wetlands was mainly determined by air humidity, indicating different responses of Collembola species to the indirect effect of warming on water availability. It is indicated that warming was the primary factor positively affecting the abundance of Collembola. An increase of Collembola abundance and community alteration to warming could have profound cascading effects on the microbes and plants they feed on in permafrost wetlands.

**Abstract:**

The consideration of environmental factors has long been crucial to developing theories about the spatial variability of species diversity. However, the effects of global warming on Collembola, in permafrost wetlands, are largely unknown. Understanding how Collembola are affected by climate warming is important as they directly affect the community assembly and decomposition processes of plant litter within soil ecosystems. A peatland area in a cold temperate monsoon climate zone in the Great Hing’an Mountains of Northeast China was selected as the study area. Collembola were captured using an aspirator after five years of simulated warming using open top chambers (OTCs). Sampling in different growth seasons showed different characteristics in the control (CK) and warming (OTCs) treatment. Further, the results showed that (1) warming treatment increased the species richness and abundance of Collembola in the different seasons, except in May, (2) warming increased Collembola abundance in permafrost wetlands, and the warming effect was more significant during the cold season (about eight times in April), (3) species composition differed significantly in the control and warming treatment in May and September, and (4) the Collembola species composition in permafrost wetlands was mainly determined by air humidity, indicating different responses of Collembola species to the indirect effect of warming on water availability. We found that warming was the primary factor positively affecting the abundance of Collembola. An increase of Collembola abundance and community alteration to warming could have profound cascading effects on the microbes and plants they feed on in permafrost wetlands.

## 1. Introduction

Permafrost wetland ecosystems are one of the most sensitive ecosystems to global warming [1]; global warming causes the degradation of permafrost, with an increase in melting depth and changes in soil moisture being observed in the permafrost regions [2]. The only zonal permafrost region in China is the Great Hing’an Mountain region. Global warming has caused severe degradation of permafrost areas in Northeast China over recent decades. The original permafrost layer is no longer “permafrost.” It has been transformed into a new active layer with seasonal melting. Furthermore, warming could increase the microbial abundance, bacterial, fungi and archaea, and greenhouse gas (CO_2_, CH_4_) emissions [3,4]. These climate change could also affect soil animals, such as Collembola, directly by changing the microenvironment in which soil animals live, or indirectly by altering resource availability and food web composition [5,6,7,8,9].

In the middle and high latitude permafrost wetlands ecosystem, microarthropods have been well studied and play a significant role in soil ecological process [10,11]. Collembola, which are one of the most important microarthropods, are widespread in wetland ecosystems and deeply involved in wetland litter decomposition and nutrient cycling [12,13]. Collembola species are widely distributed across wetlands and sensitive to wetland alterations [14]. Therefore, Collembola is a suitable taxon for evaluating the response of wetland-inhabiting soil fauna to warming. Previous studies on the effects of warming on Collembola have yielded contradictory results, in that warming was shown to exert both positive and negative effects, or insignificant effects [9,15,16]. These effects were found to be site-dependent and mainly caused by bottom-up regulation through plant and microbial changes, or the alteration of environmental factors by global warming [17,18,19]. Impacts on soil Collembola could directly affect decomposition processes within ecosystems and could disrupt secondary decomposition functions through the top-down control of bacteria, fungi, nematodes, and protozoa [20]. Therefore, it is worthwhile to study the effects of warming on Collembola in this region to improve the integrative understanding of the warming effect on animals, microbes, and soil ecological processes.

Our objectives in this study were to (i) test the effect of warming on the abundance and species richness of Collembola in the permafrost wetland in Northeast China, (ii) assess the response of community composition of Collembola to climate change by open top chamber (OTC, warming) treatment, and (iii) identify the determining factor affecting the Collembola community in the permafrost wetland.

## 2. Materials and Methods

### 2.1. Study Locality, Plots, and Experimental Design

The research station (Figure 1a,b) near the Tuqiang Forestry Bureau in the Great Hing’an Mountain, Heilongjiang Province, China, was selected as the experimental locality. This study was conducted in a natural peatland (52°44′N, 122°39′E) in the cold temperate continental monsoon climate zone, with a mean annual air temperature of −3.9 ℃ and a mean annual precipitation of 452 mm, most of which falls between July and August [21]. In late winter (15th April), the average air temperature is −2.74 ℃, the surface ice has just begun to melt, and the plants have not yet begun to germinate. In spring (23rd May), the surface ice has basically melted and the plants begin to sprout. In summer (30th August), the temperature is at the highest value for the year and the plant biomass is the largest, and in autumn (15th September), the temperature gradually decreases and the plants began to wither. *Vaccinium uliginosum* L., *Eriophorum vaginatum* L., *Chamaedaphne calyculata* L., *Moench, Sphagnum* spp., and *Ledum palustre* L. are the dominant species in this region. The soil is classified as Glacic Histoturbels according to the USDA classification system. The study area contained poor fen peatland, a sedge-dominated wetland in the continuous permafrost region. The active and transition layers were 0–60 cm and 60–80 cm below ground level, respectively, while the permafrost layer ranged 80–150 cm below the ground level. The water table depth was mostly below the surface and varied from −20 cm to −25 cm where soil samples were collected.

In the selected permafrost wetland community, homogeneous vegetation cover was selected in 2013, and 12 plots with dimensions of 100 cm × 100 cm were randomly assigned to the control and OTC treatments (six each) in the permafrost wetland to ensure equal sample size. The OTC was a square-based pyramid made of polycarbonate sheet, whose light transmittance was approximately 90%. The dimensions of the OTC included 150 cm height, 50 cm open-top quadrilateral, 45° inclination, and a 100 × 100 cm base. The wetland in the middle and high latitude permafrost area in the Great Hing’an Mountain is special, and the abundance of edaphic groups is relatively small. Study plots were situated in the continuous permafrost zone, with peat layer depths ranging from 40 to 100 cm. The surface peat layer is rich in litter, which can enhance palatability for microarthropods [22]. Therefore, we selected the ground dwelling Collembola as the research object.

OTCs significantly increased air temperature (measured in °C), which was measured at 50 cm above the soil in the control and OTC-warmed plots from 15th April to 15th September. The temperature difference between the OTC-warmed plots and the control plots was 1.23 °C on average from April to September 2018 (Table 1).

### 2.2. Sampling and Environmental Viariables

In the four periods of 2018, Collembola from the six warming and six control replicate plots were collected, using an aspirator (modified air blower, Stroke 4-GX 35, Honda, sampling diameter of opening = 14 cm) [23]. The extracted soil animals were preserved in 75% ethanol for identification. The Collembola were identified to species or morpho-species level according to the methods of Christiansen and Bellinger [24] and Hopkin [25]. Air temperature and air humidity (50 cm above the horizon) was continuously measured from 15th April to 15th September using a HOBO thermometer (Onset, Cape Cod, MA, USA) and an ECH_2_O dielectric aquameter (Em50, Digital Data Logger, Decagon Devices Inc., Pullman, WA, USA), respectively. Soil organic carbon (SOC) and soil total nitrogen (TN) were measured using an elemental analyzer (vario MACRO cube, Elementar, Langenselbold, Germany). For measuring the shoot and root biomass in control and warming treatment, the plants in the OTCs and control (length × width = 10 cm × 10 cm) plots were chose and sampled in July and September 2018. The stems of plants from both warm and control plots were cut at the ground surface and the roots and shoots were washed and dried at 80 °C for 72 h to determine their dry biomass. Soil organic carbon (SOC) and total nitrogen were also measured at the same time.

### 2.3. Data Analyses

To assess the effect of sampling time and warming treatment on species richness and abundance of Collembola, we fitted Linear mixed models (LMMs) using log-transformed response variables from the *nlme* package [26]. The first model included sampling time, warming treatment, and their interaction as the explanatory variable, with 48 plots as the random effect. Pairwise differences between different sampling times were assessed using Tukey’s honest significant difference test, and calculated using the *emmeans* package [27]. Then we applied contrasts between the warming treatment and control to estimate the effect size using the *emmeans* package. All mixed models met the assumptions of normality of residuals and homogeneity of variance.

Nonmetric multidimensional scaling (NMDS) was used to visualize the overall differences in Collembola communities at different sampling times, and was implemented in R based on species classification using the *vegan* package in R, based on the Bray–Curtis distance [27]. PERMANOVA (Permutational multivariate analysis of variance) was used to quantify differences in Collembolan community composition between warming and control treatments at different times using the *adonis* function in the vegan package.

CANOCO Version 4.5 (ter Braak and Šmilauer, 2002) was used for data analysis. A multivariate redundancy analysis (RDA) was used to analyze the effect of environmental parameters including temperature, humidity, root biomass, shoot biomass, soil organic carbon, and total nitrogen on soil Collembola communities. Collembola species with more than 10 individuals were included in subsequent analysis. Monte Carlo permutation tests (999 permutations) were used to test the significant effects.

## 3. Results

### 3.1. Environmental, Soil and Plant Properties

Root biomass increased about two times in warming treatment compared to control, but the air temperature only increased about 9.4% (18.6 ± 0.36 vs. 19.8 ± 0.35). (Table 1). Shoot biomass, soil organic carbon, C/N, and air humidity were also increased in warming treatment relative to control, but the increase was not significant.

### 3.2. Abundance and Species Richness of Collembola

We identified 33 species of Collembola belonging to 15 genera across the different times in control and warming treatments (Appendix A). Species richness and abundance of Collembola were significantly affected by sampling time (Table 2). Warming significantly increased the abundance but not species richness of Collembola (Figure 2). The abundance of Collembola was the highest in May (OTC, 14064 ± 9187) and was lower in April (660 ± 873) and September (671 ± 722) during this study. In warming treatment, Collembola abundance increased significantly in April (from 660 ± 873 to 5391 ± 5254). Species richness was higher in the warming treatment than in the control treatment in April.

### 3.3. Collembola Community Composition

Permutational multivariate analysis of variance (PERMANOVA) showed that warming treatment (*R*^2^ = 0.050, *p* < 0.001), sampling time (*R*^2^ = 0.280, *p* < 0.001), and their interaction (*R*^2^ = 0.103, *p* < 0.001) significantly affected the community composition of Collembola. In April, species composition varied marginally after the warming treatment (*R*^2^ = 0.169, *p* = 0.083), mainly due to *Sminthurinus aureus* (Lubbock, J, 1862) being more abundant in the control treatment (Figure 3). In May, species composition differed significantly in the control and warming treatment (*R*^2^ = 0.366, *p* = 0.004), mainly due to *Sinella curviseta* and *Isotoma nepalica* being dominant in warming treatments (with higher abundance). In August, species composition was similar in the control and warming treatments, but the composition was more clustered in warming treatments. In September, species composition differed significantly after the warming treatment (*R*^2^ = 0.259, *p* = 0.004), mainly due to *Protaphorura armata* and *Lepidocyrtus sp.1* being dominant in warming treatments.

### 3.4. Relationships between Collembola Communities and Environmental Factors

Changes in temperature and humidity caused by warming conditions affected the community structure of Collembola. The complete RDA model (Figure 4), including temperature, root biomass, shoot biomass, soil organic carbon, total nitrogen, and humidity explained 43% of the total variation in community composition during the OTC and CK treatments from April to September 2018 (Table 3). Collembola community structure was marginally affected by humidity (15%, *p* = 0.084). Temperature, root biomass, shoot biomass, and total nitrogen did not significantly affect the Collembola community. The abundant species, *Ceratophysella ainu*, was positively correlated with humidity, while *Lepidocyrtus felipei* was significantly affected by soil organic matter and root and shoot biomass.

## 4. Discussion

### 4.1. The Effect of Warming on Abundance and Species Richness of Collembola

Collembola community structure changes in wetlands have been well studied [28,29,30,31]. The Collembola community composition in our study was similar to that of a previous study in a wetland in Northeast China [32]. Our results found that warming significantly increased plant biomass and Collembola abundance. This is in line with the findings of former studies that found warming can promote an increase in Collembola abundance by increasing plant biomass [33]. However, some studies showed that warming could have no or negative effects on Collembola [15,16] and this adverse effect of warming on soil fauna concurrently happened when humidity declined [18,34]. While warming increased the humidity (though not significantly) in this study, humidity was not a limiting factor in the wetland in this region. Furthermore, previous studies in the same experimental area also found that warming can increase microbial biomass [3,4]. Collembola employs a r-selected life strategy and can reproduce quickly with rapidly increasing food resources [25]. The increase in microbial and plant biomass that serve as food resource of Collembola food is the main factor for the increase in Collembola in warming treatment.

### 4.2. The Effect of Warming and Sampling Time on Community Composition

Warming treatment, as well as sampling time, had a significant effect on the community composition of Collembola. Moreover, the warming treatment caused different structural changes in the Collembola community in the four sampling months. In the warming treatment, the community composition of Collembola became more aggregated (the circle of OTC became smaller), which might be due to some species (like *Sinella curviseta*, *Protaphorura armata*, and *Isotoma nepalica*) benefitting from the warming treatment, while others may disappear or respond with a decrease in numbers due to the increased dominance of certain euryok species. The result, in agreement with the results of Kustec et al., suggests that warming might affect communities in soil food webs and may impact soil ecosystem processes [35]. The warming treatment in April was not separated on Axis 1 and Axis 2. During this period, the temperature inside and outside the OTC was below zero, but because the ice had not started melting, the water available for the Collembola was limited, thereby resulting in no significant change in the community structure. Specifically, *Sminthurinus aureus* was more abundant in the control treatment; the species might be sensitive to the warming, which may be related to the growth and reproduction of the species [16]. In May, the temperature gradually increased. Because the internal temperature of OTC was higher than that of the control, the ice began to melt first, the plants germinated earlier, and the available water and food resources were higher than that of the control. Therefore, the community structure significantly differed in spring because of warming. In August, species composition was more clustered in warming treatments, the Collembola community structure was relatively the same in the warming and control treatment and the temperature advantage of the warming treatment was no longer evident. However, in September, species composition differed significantly after the warming treatment, mainly due to *Protaphorura armata* being dominant in warming treatments. The community structure results of different seasons showed that environmental factors and food resources might jointly affect the changes in the Collembola community structure, and the temperature became the main factor in the early stage of freezing and thawing. We concluded that the warming treatment influenced the Collembola community structure changes in each growing season, but the characteristics were different in the four sampling periods.

### 4.3. Relationship with Environmental Factors

The trend of the effect of warming on abundance and species richness of Collembola is obviously inconsistent. The response of Collembola has been reported to be negative [36,37], positive [38,39], or neutral [40,41]. OTC warming treatment decreases soil moisture [37,42,43], and Collembola are considered to be sensitive to low soil moisture level. Therefore, in the OTC warming experiment of ecosystems with soil moisture as the limiting factor, Collembola are negatively affected [36,44]; additionally, it has a negative impact on the biomass and diversity of Mesofauna in arid areas, but shows a positive impact on the biomass and diversity of mesofauna in humid areas [45]. Most studies suggest that soil fauna are bottom-up controlled through food resource alteration [46,47]. In mid–high latitude wetlands of our study region, warming increases water content due to earlier thawing in late winter and later freezing in early winter. Our RDA result also showed that humidity was the most important factor which determined Collembola community structure, and the most abundant species, *Ceratophysella ainu*, was positively correlated with humidity. It was indicated that soil humidity variation in the warming treatment could increase the fitness of Collembola.

## 5. Conclusions

Although some studies have focused on the effects of warming on soil fauna, the response of soil fauna in high-latitude wetlands to warming is still lacking [48]. Such research is significant because soil fauna control many soil processes and are connected to aboveground vegetation. Due to a lack of macro soil fauna, microarthropods such as Collembola play a particularly important role in the wetland ecosystem in the middle and high latitudes.

The findings of this study indicate that warming increases the Collembola abundance in permafrost wetlands (about eight times in April), and that the warming effect was most evident at colder temperatures. Local heterogeneity may be very important for soil fauna composition and may play a key role in buffering the impact of future climate change on Collembola. Warming combined with higher humidity and food resources, such as shoot and root biomass, increased the fitness of Collembola in the permafrost wetland in Tuqiang, Northeast China. Further research is needed on the relationships between warming, soil fauna, microbes, and soil processes to advance our understanding on how future climate change will affect soil communities.

## Figures and Tables

**Figure 1 insects-14-00033-f001:**
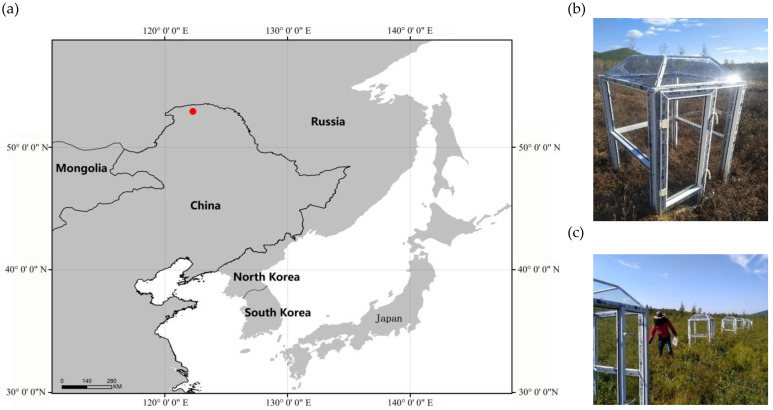
(**a**) Study area in the middle and high latitude permafrost area in the Great Hing’an Mountain, Heilongjiang Province, China. (**b**) Open top chambers (OTCs) installed in the study area.

**Figure 2 insects-14-00033-f002:**
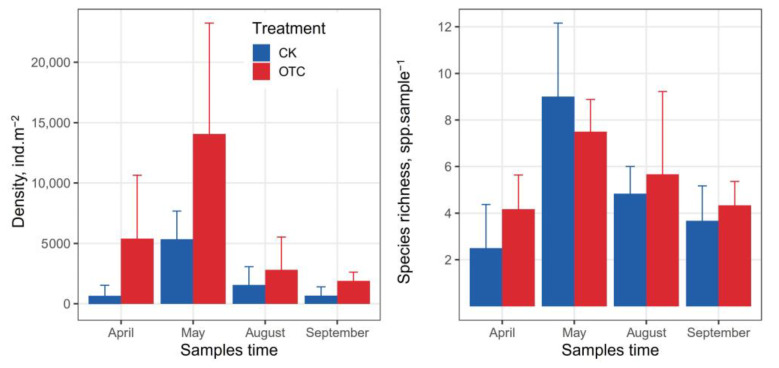
Box plots of abundance and species richness of Collembola in April, May, August, and September at the permafrost wetland community sites and in the control (CK) and warming treatment (OTC) at Tuqiang field station, Great Hing’an Mountain. Bars with the same upper letter do not differ significantly (*p* > 0.05).

**Figure 3 insects-14-00033-f003:**
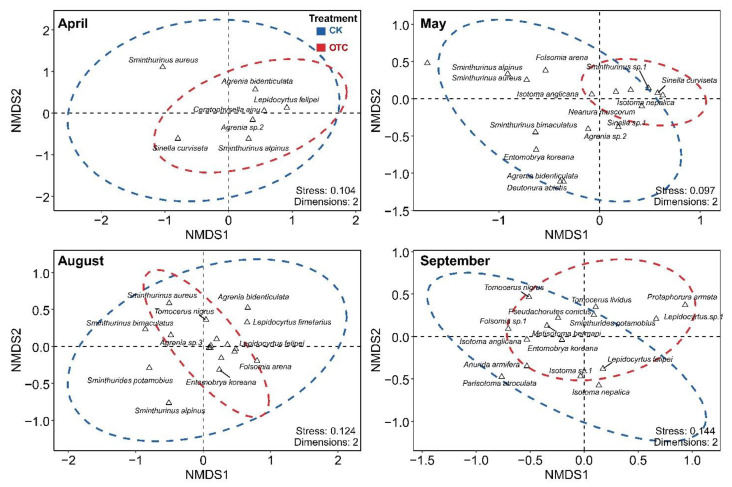
Non-metric multidimensional scaling (NMDS) ordinations of the Collembola communities at different sampling time (April, May, August, and September) and treatment (CK, OTC).

**Figure 4 insects-14-00033-f004:**
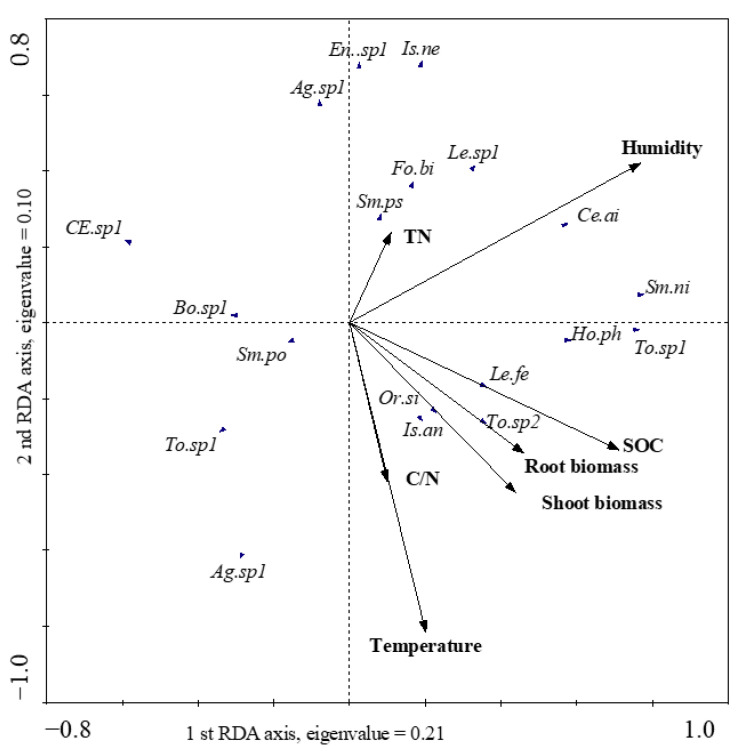
Ordination biplot of the redundancy analysis (RDA) with mean abundance of Collembola species under open top chamber (OTC) treatment and soil properties in the Tuqiang wetland of Northeast China. For the full names of the Collembola see Appendix A.

**Table 1 insects-14-00033-t001:** The values (mean ± SE, *n* = 6) of environmental factors in control and warming treatment in CK and OTC treatment (CK = Control, OTC = Open top chamber). The different letters next to each value represent significant differences (*p* < 0.05).

Samples	Shoot Biomass	Root Biomass	Soil Nitrogen	Soil Organic Carbon	C/N	Air Humidity (%)	Air Temperature (℃)
CK	752.7 ± 58.03 ^a^	70.37 ± 21.22 ^b^	6.27 ± 1.72 ^a^	487.77 ± 29.93 ^a^	85.08 ± 28.98 ^a^	13.02 ± 3.48 ^a^	18.6 ± 0.36 ^b^
OTC	965.47 ± 425.8 ^a^	144.17 ± 39.85 ^a^	5.2 ± 2.2 ^a^	537.27 ± 73.87 ^a^	112.77 ± 34.78 ^a^	17.29 ± 6.71 ^a^	19.8 ± 0.35 ^a^

**Table 2 insects-14-00033-t002:** F- and *p*-values of linear mixed-effects models on the effect of warming treatment, sampling time and their interactions on the abundance and species richness (the number of species per sample) of Collembola in the Tuqiang wetland of Northeast China. *p*-values < 0.05 are interpreted as significant and given in bold.

Variables	Abundance of Collembola			Richness of Collembola		
	Df	F-Value	*p*	Df	F-Value	*p*
Warming	1.10	18.6619	0.002	1.10	1.6323	0.23
Time	3.30	11.6548	<0.001	3.30	13.5216	<0.001
Warming × Time	3.30	2.0633	0.126	3.30	2.105	0.121

**Table 3 insects-14-00033-t003:** Redundancy analysis of soil property effects on Collembola community composition in warming (open top chamber, OTC) and control (CK) treatments from April to September in the Tuqiang wetland of Northeast China. The marginal effects (i.e., effects when the particular terms were used as the only explanatory variable), and the total amount of explained variation (%) are listed. *p*-values are based on Monte Carlo permutation tests.

Environmental Variables	%	F	*p*
Humidity	15	1.78	0.084
Temperature	13	1.62	0.116
Root biomass	8	0.94	0.508
Shoot biomass	3	0.45	0.880
Total nitrogen	4	0.42	0.870
Total % variance explained	43	——	——

## Data Availability

Data available on request from the authors.

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
