# Peer review of "Warming in Cold Seasons Increases the Abundance of Ground-Dwelling Collembola in Permafrost Wetlands"

_insects, 2022, doi:10.3390/insects14010033_

Round 1

Reviewer 1 Report (New Reviewer)

This study describes how the collembola community of a wetland is affected by a field climate warming simulation in China. I like the fact that multiple samples were taken during the year. The experimental set-up is straightforward, although the size and shape of the OTCs are unusual compared to the ITEX ones and any impacts this has on the enclosed microclimate  could perhaps be explained in more detail. Were they left in place year-round? if so, what happened with any snow which would have accumulated considerably within these tall frames.

The statistical approach is incorrect as sampling through time is done on the same experimental plots which violates independence of samples. This makes the current statistical output unreliable.

The use of control and warming as explanatory factors in the RDA plots is to my  mind incorrect (see comments below) and this analyses should be re-done.

I would like to see more explanation about the extraction method used. Most soil community data is based on soil samples and the use of an aspirator may greatly affect the outcome.

The use of air temperature measured at 50 cm seems inappropriate for organisms living on-in the soil.

The results/discussion remains quite superficial. Although an ordination is made (which requires redoing) nothing much is made of any effect of the changes in shoot and root biomass (or plant composition altogether) and how this could affect the collembola community.   

Lines 122-124 what was the impact of the OTCs on soil temperature? I would think that is of greater importance for soil dwelling organisms than that of the air above them.

Line 127; an aspirator is a simple air sucking device. How do you ensure that the time and surface area sampled are equal between experimental plots? How can you be sure that animals did not move away during sampling?

Line 130 temperature of what?

Line 131 humidity was measured of what?

Line 132 C and N were analysed of what kind of substrate/tissue?

Line 134: what do the measurements between brackets indicate?

Line 144 considering that the ‘time’ samples were taken from the same experimental plots these are not independent samples (one of the pre-requisites of an ANOVA) which requires a mixed effect model approach (or repeated measures).

Lines 178-181; the ANOVA results are quite simple (although faulty for the type of data collected; see comment above); with a warming and time effect; the lack of a significant interaction indicates that warming was consistent across all sampling times. The comment that warming was non-significant for certain measuring periods does not follow from the ANOVA table for any of the measured Collembola variables.

Lines 183-184 There was a time effect but no warming effect so please remove ‘control’ from this sentence (please keep in mind that the ANOVA approach is not valid here).

Figure 2; please include units with the y-axis. Considering the large variation between treatments and plots was the data normally distributed? Did you test for this and/or apply transformations? The Tukey lettering is incorrect given the ANOVA results (you cannot distinguish between time-warming because there was no significant interaction)

Line 193 I can see how during spring CK and OTC are somewhat separated but this is not clear for summer (3c). The data in figure 3d (autumn) do not suggest that there is a significant difference at all between treatments. Please have a look at this part again.

Lines 205-206 please first explain these temperature and humidity response to the warming treatments. What evidence is there that these two factors were the cause for any observed differences in collembola?

Figure 4 you cannot include the ‘control’ and ‘warming’ treatments as variables within these types of ordinations. The strong ordination along the first axis with these two variables simply results from the fact that there are 6 control and 6 warming plots; these variables therefore have the largest spread in variation but that is simply because you called these plots ‘control’ and ‘warming’ and this has nothing to do with the biological and soil variables measured. ‘Temperature’ already takes the warming effect into account here. So please redo this analyses and leave out the ‘control’ and ‘warming’

Lines 229-231 I follow the first part of the sentence but it is unclear how the conclusion is drawn here. Where is the evidence that the food availability changes during May?

Lines 232-234 if these are re-emerging collembola their reproductive output is of little relevance here.  Also there is an unspoken assumption that predators also do not re-emerge. Couldn’t their predators not survive winter and start hunting during spring? Any of the statements made in these lines needs backing up with references that show that these mechanisms are in play here.

Ilne 236 change in relation to what/during which period?

Lines 238 was wind affected by the OTCs? Please explain intensity change and how/when. Also by how much was the soil temperature affected by OTCs? How does this relate to freezing-thawing and soil degree day sums (thermal energy)

Lines 239-249 apart from the fact that the statistics for these results are incorrect; there is a lot of speculation here that is not supported by the presented data. No evidence is provided for increased primary productivity and or how any of this is directly associated with the observed Collembola community patterns.

Line 260 please provide evidence of this earlier melt out.

Considering that para 4.2 is on the community composition I was surprised that species changes (or even loss/gains of species) are not described at all. Which species benefit the most and which are in decline? Are the commonalities among the species that change? Can you speculate briefly what the consequences of these change may be for ecosystem processes and/or other organisms within this ecosystem?

4.3. Relationship with environmental factors

Considering that this para deals with the abiotic conditions within the control and OTCs it would have been great if temperature and water availability are placed into context. The results section does not provide any handholds as to by how much temperature/moisture are affected by the treatments. The discussion on how local moisture can influence collembola communities to warming is meaningful but only if the information of the current study on moisture and temperature effects are included.

282-284 I would prefer to see a figure or table that shows by how much soil moisture was affected by the treatments.

Lines 286-289 please reconsider these lines as there are numerous studies on climate change impact on soil biota.

 Line 296 no evidence is provided that temperature, moisture and food availability went up under warming!

Author Response

Reviewer 1

Comment 1 (abbreviated as C1): This study describes how the collembola community of a wetland is affected by a field climate warming simulation in China. I like the fact that multiple samples were taken during the year. The experimental set-up is straightforward, although the size and shape of the OTCs are unusual compared to the ITEX ones and any impacts this has on the enclosed microclimate could perhaps be explained in more detail. Were they left in place year-round? if so, what happened with any snow which would have accumulated considerably within these tall frames.

Response 1 (abbreviated as R1): Thanks a lot for pointing this out. As shown in Figure 1b, there is an inclined angle above our OTC, and snow is not easy to stay on it. The study field is quite open and windy, so there is basically no accumulation of snow at the top of the OTC.

C2: The statistical approach is incorrect as sampling through time is done on the same experimental plots which violates independence of samples. This makes the current statistical output unreliable.

R2: Thank you for pointing this out. We have, accordingly, fitted linear mixed-effect models that included 48 plots as the random effect. please see line 132-134.

C3: The use of control and warming as explanatory factors in the RDA plots is to my mind incorrect (see comments below) and this analyses should be re-done.

R3: Done, we have redone the RDA. Please see Figure 5 and table 3.

C4: I would like to see more explanation about the extraction method used. Most soil community data is based on soil samples and the use of an aspirator may greatly affect the outcome. [Method]

R4: Done. Aspirator was used in our experiment. The front end of the aspirator is a square iron frame (modified air blower, Stroke 4-GX 35, Honda, sampling diameter of opening = 14 cm). The staying time of the aspirator has been investigated in the former studies, which can ensure the absorption of the springtails below the iron frame area. Our study area located in a wetland with a certain degree of water accumulation on the surface, mainly composed of ground-dwelling springtails with strong mobility. The soil drills collected mainly euedaphic groups, so we chose to use the aspirator to collect the Collembola groups that appeared in this experiment. For details, please see attachment.

C5: The use of air temperature measured at 50 cm seems inappropriate for organisms living on-in the soil. [Method]

R5: Done. Open-top chambers were used to study the impact of warming on wetland ecosystems. Warming mainly affects springtails though wetland plants; and the impact of warming on plants is mainly through the impact of air temperature. Therefore, we chose to investigate the temperature that is 50cm above the ground.

C6: The results/discussion remains quite superficial. Although an ordination is made (which requires redoing) nothing much is made of any effect of the changes in shoot and root biomass (or plant composition altogether) and how this could affect the collembola community. [Discussion]

R6: Done, the statistical analysis, results, discussion and RDA were changed as the reviewer suggested, please see the corresponding section.

C7: Lines 122-124 what was the impact of the OTCs on soil temperature? I would think that is of greater importance for soil dwelling organisms than that of the air above them. [Method]

R7: Done, the research field of this experiment is a wetland, with a certain amount of accumulated water on the surface, and little "soil", which is basically composed of plant roots. The effect of warming on the temperature of "soil" should be the effect of warming on water. This could buffer the effect of warming. Therefore, we tested the temperature that has a greater impact on wetland plants to reflect the effect of warming. Moreover, the effect of warming on ground-dwelling springtails were mainly caused by the change of plant properties in wetlands.

C8: Line 127; an aspirator is a simple air sucking device. How do you ensure that the time and surface area sampled are equal between experimental plots? How can you be sure that animals did not move away during sampling? [Method]

R8: Done. Aspirator was used in our experiment. The front end of the aspirator is a square iron frame (modified air blower, Stroke 4-GX 35, Honda, sampling diameter of opening = 14 cm). The staying time of the aspirator has been investigated in the former studies, which can ensure the absorption of the springtails below the iron frame area. Our study area located in a wetland with a certain degree of water accumulation on the surface, mainly composed of ground-dwelling springtails with strong mobility. The soil drills collected mainly euedaphic groups, so we chose to use the aspirator to collect the Collembola groups that appeared in this experiment.

C9: Line 130 temperature of what? Of soil or litter? [Method]

R9: Done, we revised the text as you suggested. Please see line 120-123.

C10: Line 131 humidity was measured of what? [Method] Humidity of soil?

R10: Done, we revised the text as you suggested. Please see line 120-123.

C11: Line 132 C and N were analysed of what kind of substrate/tissue? [Method]

R11: Done, we revised the text as you suggested. Please see line 129-130.

C12: Line 134: what do the measurements between brackets indicate?

R12: Done, for measuring the shoot and root biomass in control and warming treatment, the plants in the OTCs and control (length × width = 10 cm × 10 cm) plots were chose and sampled in July and September 2018. Please see line 125-129.

C13: Line 144 considering that the ‘time’ samples were taken from the same experimental plots these are not independent samples (one of the pre-requisites of an ANOVA) which requires a mixed effect model approach (or repeated measures).

R13: We have redone the analysis using linear mixed-effect model and included the plots as the random effect. Please see line 132-140.

C14: Lines 178-181; the ANOVA results are quite simple (although faulty for the type of data collected; see comment above); with a warming and time effect; the lack of a significant interaction indicates that warming was consistent across all sampling times. The comment that warming was non-significant for certain measuring periods does not follow from the ANOVA table for any of the measured Collembola variables.

R14: We have redone the analysis using linear mixed-effect model and included the plots as the random effect and rewrote the result accordingly. “Warming treatment and sampling time significantly affected species abundance of Collembola (Table 2). The abundance of Collembola was the highest in May and was lower in April and September during this study (Fig. 2). After warming treatment, Collembola abundance increased significantly in April.” Please see line 164-170.

C15: Lines 183-184 There was a time effect but no warming effect so please remove ‘control’ from this sentence (please keep in mind that the ANOVA approach is not valid here).

R15: Done. We have redone the analysis using linear mixed-effect model and included the plots as the random effect and rewrote the result accordingly. Please see line 132-140.

C16: Figure 2; please include units with the y-axis. Considering the large variation between treatments and plots was the data normally distributed? Did you test for this and/or apply transformations? The Tukey lettering is incorrect given the ANOVA results (you cannot distinguish between time-warming because there was no significant interaction)

R16: Thank you for pointing this out. We have, accordingly, fitted linear mixed-effect models that included 48 plots as the random effect. Then rephrased the result accordingly. “To assess the effect of sampling time and warming treatment on species richness and abundance of Collembola, we fitted Linear mixed models (LMMs) using log-transformed response variables using nlme package (Pinheiro, 2022).” Please see line 132-140.

C17: Line 193 I can see how during spring CK and OTC are somewhat separated but this is not clear for summer (3c). The data in figure 3d (autumn) do not suggest that there is a significant difference at all between treatments. Please have a look at this part again. [NMDS]

R17: Thank you for pointing this out. We have rephrased the results. “In September, species composition differed significantly after the warming treatment (R2 = 0.259, p = 0.004), it was mainly due to Protaphorura armata (Tullberg, 1869) and Lepidocyrtus sp.1 were dominant after warming treatments.” Please see line 198-200.

C18: Lines 205-206 please first explain these temperature and humidity response to the warming treatments. What evidence is there that these two factors were the cause for any observed differences in Collembola?

R18: Done, we have supplemented “Table 1” to show the environmental and plant properties in OTC and control treatment. Please see line 155-157.

C19: Figure 4 you cannot include the ‘control’ and ‘warming’ treatments as variables within these types of ordinations. The strong ordination along the first axis with these two variables simply results from the fact that there are 6 control and 6 warming plots; these variables therefore have the largest spread in variation but that is simply because you called these plots ‘control’ and ‘warming’ and this has nothing to do with the biological and soil variables measured. ‘Temperature’ already takes the warming effect into account here. So please redo this analyses and leave out the ‘control’ and ‘warming’ [RDA]

R19: Thanks a lot for your nice comment. Done, we have redone the RDA. Please see Figure 5 and table 3.

C20: Lines 229-231 I follow the first part of the sentence but it is unclear how the conclusion is drawn here. Where is the evidence that the food availability changes during May? [Discussion]

R20: Done, we revised the text as you suggested. Please see line 238-245.

C21: Lines 232-234 if these are re-emerging collembola their reproductive output is of little relevance here. Also there is an unspoken assumption that predators also do not re-emerge. Couldn’t their predators not survive winter and start hunting during spring? Any of the statements made in these lines needs backing up with references that show that these mechanisms are in play here. [Discussion]

R21: Done, Thanks a lot for your suggestion. This sentence really lacks clear evidence and references to support it, it just lists the possibilities, so we decide to delete this sentence.

C22: line 236 change in relation to what/during which period? [Discussion]

R22: Done, Thanks a lot for your suggestion. We revised the text as you suggested. Please see line 234-235.

C23: Lines 238 was wind affected by the OTCs? Please explain intensity change and how/when. Also by how much was the soil temperature affected by OTCs? How does this relate to freezing-thawing and soil degree day sums (thermal energy) [Discussion]

R23: Done, Thanks a lot for your suggestion. OTC did reduce wind strength, but we didn't measure it. It is just a simple guess. Therefore, we decided to focus on the overall warming effect of OTC, and put the results of temperature and humidity alteration (from April 15th to September 15th) in Table 1. Please see line 155-157.

C24: Lines 239-249 apart from the fact that the statistics for these results are incorrect; there is a lot of speculation here that is not supported by the presented data. No evidence is provided for increased primary productivity and or how any of this is directly associated with the observed Collembola community patterns.

R24: Done, Thanks a lot for your suggestion. We have revised the data analysis and results in the text. The text in Lines 239-249 were also revised, please see line 238-245.

C25: Line 260 please provide evidence of this earlier melt out.

R25:Thanks a lot for your nice comment. Done, we have supplemented Table 1 to show the temperature increasing in OTC treatment. Please see line 155-157.

C26: Considering that para 4.2 is on the community composition I was surprised that species changes (or even loss/gains of species) are not described at all. Which species benefit the most and which are in decline? Are the commonalities among the species that change? Can you speculate briefly what the consequences of these change may be for ecosystem processes and/or other organisms within this ecosystem?

R26: Thank you for pointing this out. We have rephrased the results give more information about species. Please see line 247-273.

C27: 4.3. Relationship with environmental factors Considering that this para deals with the abiotic conditions within the control and OTCs it would have been great if temperature and water availability are placed into context. The results section does not provide any handholds as to by how much temperature/moisture are affected by the treatments. The discussion on how local moisture can influence collembola communities to warming is meaningful but only if the information of the current study on moisture and temperature effects are included.

R27:Thanks a lot for your nice comment. Done, we have supplemented Table 1 to show the temperature increasing in OTC treatment. Please see line 155-157. We believe that the increase in humidity caused by warming is mainly to increase the biomass of wetland plants. As a food resource for collembola, the increase of plant biomass will promote the increase of Collembola abundance. Please see line 285-288.

C28: 282-284 I would prefer to see a figure or table that shows by how much soil moisture was affected by the treatments. [RDA]

R28:Thanks a lot for your nice comment. Done, we have supplemented Table 1 to show the temperature and humidity increasing in OTC treatment. Please see line 155-157.

C29: Lines 286-289 please reconsider these lines as there are numerous studies on climate change impact on soil biota.

R29:Done, we revised the text as you suggested. Please see line 290-291.

C30: Line 296 no evidence is provided that temperature, moisture and food availability went up under warming!

R30: Done, we have supplemented “Table 1” to show the environmental and plant properties in OTC and control treatment. Please see line 155-157.

Reviewer 2 Report (New Reviewer)

Overview

This study examines the effects of increased temperature on Collembola assemblages in a permafrost wetland in northeast China.  The authors performed an in-situ experiment that involved control and treatment chambers.   At the time of this study (2018), temperature was increased 1.3 degree relative to the control treatments. Collembola assemblages, as well as their food resources and other environmental variables, were sampled in four seasons in 2018. This study has implications for the effects of climate change on arthropod communities in permafrost regions and is well within the scope of the journal Insects. 

General comments

In general, the manuscript is clear and well-written but there are areas that could be reorganized to improve clarity and redundancy.  More specifically, the ANOVA results and the first paragraph of the Discussion.  Lines 56-62 should be moved to the Introduction and line 119 could also be moved to the introduction. 

The methods related to collembola sampling need more detail.  This is experiment cannot be repeated as the method are currently described on line 126.  I assume sampling with the aspirator was standardized, but how? By a standard area within the plot?  By time and effort?  Along the same lines, what is mean by use of the term ‘activity density’ as the response variable throughout the manuscript?  Activity and density are probably correlated but they are not the same response.  There is no description of how activity was measured. If collembola were truly collected per unit area then ‘individuals per unit area’ should be reported, which is a true measure of density. This issue can be resolved simply by describing sampling methods in detail and then eliminating the word ‘activity’.

The environmental data is not shown in the manuscript.  The authors need to show the environmental data (the range, mean or median) in the main body of the manuscript.  This is important because other researchers will be interested in the true values for comparisons to other studies.  Temperature data in the appendix should be included in this table, rather than in the appendix, especially since this is the main variable of interest.  Additionally, the chart in Appendix A needs more detail in the caption.  Is each point a year or sampling times?  Collembola were sampled only 4 times during the last year so why are there more than four points in each month ? 

Why were four separate NMS ordinations (one for each season) performed ?  One NMS should capture the pattern reported but would also show variation among the seasons.

Please explain why ‘warming’ and ‘control’ labels were included as variables in the RDA?  I view these as unnecessary, categorical variables that could result in overfitting the model. This is also related to the ‘warming’ variable indicated on line 206. 

Line 31.  Add an ‘s’ to season

Line 58.  Collembola is are

Line 80. study were that to

Line 134.  Plant sampling needs clarification.  The way the methods are currently written suggest that the OTCs and control were 10 by 10 cm.  I assume the plan sampling plots were 10 by 10 cm and were within the OTC chambers which were 100 by 100 cm. 

Line 192.  Is the term ‘beta-diveristy’ necessary.  This is confusing because diversity was not calculated.  Perhaps the term ‘composition’ would be better here

Line 231.  The point of this sentence is not clear.  Rewrite for clarification. 

Line 268.  Capitalize ‘we’

Line 296.  Please expand on ‘food resources.’ What about food resources?  Were the food resources of higher or lower quality (based on C/N ratios) and quantity?  This is a nice opportunity to discuss the C/N values from treatment chambers.

Author Response

Reviewer 2

General comment

Dear Editor and authors,

C31: This paper about effect of warming on soil Collembola in such unique ecosystem, permafrost wetland, is highly interesting, and this work is an important contribution to previous projects and publications focused on global warming and soil fauna. According to the authors, results of these studies might be used to predict consequences of the current climate warming for ecosystem function. In present, I see several deficiencies. The most important are listed below. There is a need to verify some references in the text of MS. It is possible, that they moved? Please, check them carefully. In „Discussion“ there is a need to delete all Tables and Figures which are mentioned in this text, it is not necessary. In subsection 4.2., there are also confusions (Fig. 3a-d instead of Fig. 2a-d). Note that, as far as I'm not a native speaker, I have not evaluated the grammar nor the general writing style.

(I follow lines in Pdf MS)

R31: Done, thanks a lot for your comments. We have checked all the references in the text. properties in OTC and control treatment. We also checked the figure number in the text. Please see the corresponding section.

C32: Introduction

Line 46. „on the effect of climate change on plant and soil microbes“, this is not written correctly in terms of references used, soil microarthropods/arthropods instead of soil microbes.

R32: Done. Thanks a lot for your comment. I think it is right here. We want to emphasize that most of the current research focuses on microorganisms and plants, and less research on soil animals.

C33: Line 51. There is again used „play ...role“ as repetition with previous sentence. Authors can reformulate it, e.g. „Collembola, widespread in wetland ecosystems, have an important role in wetland litter decomposition and soil nutrient cycling.“

R33: Done. We revised the text as reviewer suggested, please see line 51.

C34: Lines 55-56. In this parahraph authors deal with wetlands ecosystems and these two sentences are out of the context: „Collembola can also successfully survive in high latitude Arctic regions. In most habitats, the abundance and species richness of terrestrial arthropods exceed that of all other phyla“ Please, delete or  make this part more clearly for reader.

R34: Done. We revised the text as reviewer suggested, we delete this irrelevant part.

C35: Lines 68-69. „Such changes impact humidity, soil nutrients, soil microbes, and soil Collembola [22]“ It is unclear how can these changes affect soil Collembola in terms of reference used, this refrence informs us about importance of microbial abundance related to soil nutrients. I suggest to remove „soil Collembola“ from this sentence or to use relevant reference for this idea.

R35: Done. We revised the text as reviewer suggested, please see line 67.

C36: Lines 77-79. There is redundant text, repetitions with Methods section, the same data are presented in Methods section. I suggest to remove these sentences in this part.

R36: Done. We revised the text as reviewer suggested, and this part was deleted.

C37: Line 80. It is unclear how was defined parameter activity-density for Collembola communities, and also how was calculated this parameter for Collembola communities which were collected by aspirator. In ecological studies, specialists always use basic quantitative community parameters for soil fauna such as abundance in the case of soil samples/gradient extraction technique, activity/activity-densities in the case of pitfall traps/trap samples, also term density is often used in context of populations. Can authors clarify and specify it in „Methods section?

R37: Done. We revised the text as reviewer suggested, we replace the word “activity-density” with “abundance”, please see line 72.

C38: Line 81. Beta or alpha diversity of soil Collembola communities? Also inconsistent in the text of MS. Please, clearly specify second aim of your study.

R38: Done. We replace “Beta or alpha diversity” with “Community composition”, please see line 74.

C39: Materials

For this subsection I suggest title „Study locality, plots and experimental design“

Line 87. (Fig. 1a,b) instead of (Fig. 1)

R39: Done. We revised the text as reviewer suggested, please see line 79.

C40: Line 88. experimental locality instead of experimental site

R40: Done. We revised the text as reviewer suggested, please see line 78.

C41: Line 88-97. I especially like the very accurate background informations on the research locality/plots, sampling design and environmental measurements, there is important to note more details for readers in section „Methods“. The description of methods section needs to be expanded. Please, give reference for regional climatic data of study area (temperature and precipitation). It is unclear which variables relate to air and which relate to soil (e.g., temperature can be related to both) in the text of ms. This should be clearly specified. „The average temperature -2.74 ℃“... own temperature measurements, continually measured soil or air temperature in 2018, or literary source? How was recorded vegetation structure in study locality (specialist, literary source...)?

R41: Done. We revised the text as reviewer suggested, please see line 80-90.

C42: Line 115. Study plots instead of study sites.

R42: Done. We revised the text as reviewer suggested, please see line 107.

C43: For this subsection I suggest title „Sampling and environmental viariables“

R43: Done. We revised the text as reviewer suggested, please see line 115.

C44: Lines 126-127.  Four sampling months/periods is more precisely than four sampling seasons, but I leave it on authors decision. Is Zhang et al. 2018 correct reference for this part?

R44: Done. We revised the text as reviewer suggested, please see line 116-118. We replace “Zhang et al., 2018” with “Zou et al., 2016” which is the earliest description of this “modified leaf blower”. Please see line 116-118.

C45: Line 128. extracted individuals instead of organisms

R45: Done. We replace “organisms” with “soil animals” , please see line 118.

C46: Line 130 and 137. How was measured air temperature by thermometer/data-loggers in the OTCs and control plots from April to September? Continually? Are there single measurements for soil humidity and others edaphic factors? Please, specify it more precisely. [Method]

R46: Done. We revised the text as reviewer suggested, please see line 120-123.

C47: I suggest soil organic carbon (SOC) instead of SOC.

R47: Done. We revised the text as reviewer suggested, please see line 123.

C48: Data analyses

Line 144. activity-density vs. abundance. The use of term abundance for Collembola communities in the text of ms is more appropriate than activity-density, please, see also my comment above.

R48: Done. We revised the text as reviewer suggested, please see line 133.

C49: Line 156. Collembola communities instead of Collembola community.

R49: Done. We revised the text as reviewer suggested, please see line 150.

C50: Results

Why are basic data such as soil humid and soil-chemical data (SOC, NT, C/N) for individual plots/treatments not recorded in “Results”. Authors use them in analyses and in “Discussion” (Lines 229-231) they outline their importance, explain the higher spring community abundances of Collembola in terms of edaphic factors. Please, I advise to add these useful data for readers.

R50: Done. We revised the text as reviewer suggested, environmental factors, soil and plant properties were included in results, please see line 155-157.

C51: For this subsection I suggest title „Collembola abundance and diversity“

R51: Done. We revised the text as “abundance and species richness of Collembola”, please see line 162.

C52: Line 161. We identified 33 species of Collembola belonging to 15 genera across the different climate treatments. Please, here you can add (Appendix B).

R52: Done. We revised the text as reviewer suggested, please see line 156.

C53: Lines 162-166. Please, delete redundant species authority for each species in this part, they are recorded in Appendix B.

R53: Done. We delete this sentence. It is not a useful description of the result.

C54: Line 184. Please, remove here redundant Fig. 2b.

R54: Done. We revised the text as reviewer suggested, please see line 170.

C55: Line 192. Diversity instead of beta diversity.

“Collembola community composition”

R55: Done. We replace “beta diversity” with “community composition”, please see line 191.

C56: Line 206- 210. How many species were included in RDA analysis and what criterion was used to select species for this analysis? Please, give this information. Soil organic carbon instead of soil organic matter and also total nitrogen instead of NT. [RDA]

R56: Done. We revised the text as reviewer suggested, we supplemented the criterion which was used to select species for RDA, please see line 150-151. Soil organic carbon and total nitrogen were also revised, please see line 207.

C57: Line 209. Please, remove redundant „in the Tuqiang wetland in Northeast China“

R57: Done. We revised the text as reviewer suggested, please see line 209.

C58: Lines 218-219. Please, reformulate this sentence in Figure 4 more precisely, e.g. for the full names of the Collembola taxa see Appendix B.

R58: Done. We revised the text as reviewer suggested, please see line 216-217.

C59: Appendix B: Please, add code/abbreviations of Collembola species in Appendix B

R59: Done. We revised the text as reviewer suggested, please see Table A1.

C60: Lines 212- 213. “Lepidocyrtus felipei was significantly affected by soil organic matter and root and shoot biomass” ...by root biomass, is it correct? Similarly, it is unclear for me that “Ceratophysella ainu was positively correlated with humidity “

R60: Thanks a lot for your nice comment. Done, we have redone the RDA. It is right in the new figure. Please see fig. 5.

Discusion

C61: For first subsection I suggest title “The effect of warming on Collembola abundance and diversity”. Please, include also critical discussion about effect of warming on Collembola diversity (literary data and results of our study).

R61: Done. We have revised this subsection to focus on Collembola abundances with significant changes when encountered warming, please see line 227.

C62: Lines 226-231. There is unclear first part of paragraph for readers, please reformulate it (what Collembola community structure changes..., previous study focused on...etc.).

R62: Done. We revised the text as reviewer suggested, please see line 228-231.

C63: Line 241. Remove redundant (Table 1), similarly Fig. 2 and Table 2 (Lines 250-251).

R63: Done. We revised the text as reviewer suggested, please see line 242.

C64: Line 252. the effect of warming instead of the effect of species warming

R64: Done. We revised the text as reviewer suggested, please see line 246.

Conclusions

C65: Lines 288. „..long-term research on underground biota is needed”... I would like to advise to add relevant and recent reference in this context, useful for both authors and readers:

R65: Done. We revised the text as reviewer suggested, please see line 290-291.

C66: Appendix A. Please, add unit for Temperature (axis Y). More precisely, there is average temperature. “The average temperature of different treatments at different sampling times”

R66: Done. We make a table to show the environmental, soil and plant properties in OTC and control treatment, please see table 1.

Reviewer 3 Report (New Reviewer)

REF#1

General comment

Dear Editor and authors,

This paper about effect of warming on soil Collembola in such unique ecosystem, permafrost wetland, is hihly interesting, and this work is an important contribution to previous projects and publications focused on global warming and soil fauna. According to the authors, results of these studies might be used to predict consequences of the current climate warming for ecosystem function. In present, I see several deficiencies. The most important are listed below. There is a need to verify some references in the text of MS. It is possible, that they moved? Please, check them carefully. In „Discussion“ there is a need to delete all Tables and Figures which are mentioned in this text, it is not necessary. In subsection 4.2., there are also confusions (Fig. 3a-d instead of Fig. 2a-d). Note that, as far as I'm not a native speaker, I have not evaluated the grammar nor the general writing style.

(I folow lines in Pdf MS)

Introduction

Line 46. „on the effect of climate change on plant and soil microbes“, this is not written correctly in terms of references used, soil microarthropods/arthropods instead of soil microbes.

Line 51. There is again used „play ...role“ as repetition with previous sentence. Authors can reformulate it, e.g. „Collembola, widespread in wetland ecosystems, have an important role in wetland litter decomposition and soil nutrient cycling.“

Lines 55-56. In this parahraph authors deal with wetlands ecosystems and these two sentences are out of the context: „Collembola can also successfully survive in high latitude Arctic regions. In most habitats, the abundance and species richness of terrestrial arthropods exceed that of all other phyla“ Please, delete or  make this part more clearly for reader.

Lines 68-69. „Such changes impact humidity, soil nutrients, soil microbes, and soil Collembola [22]“ It is unclear how can these changes affect soil Collembola in terms of reference used, this refrence informs us about importance of microbial abundance related to soil nutrients. I suggest to remove „soil Collembola“ from this sentence or to use relevant reference for this idea.

Lines 77-79. There is redundant text, repetitions with Methods section, the same data are presented in Methods section. I suggest to remove these sentences in this part.

Line 80. It is unclear how was defined parameter activity-density for Collembola communities, and also how was calculated this parameter for Collembola communities which were collected by aspirator. In ecological studies, specialists  always use basic quantitative community parameters for soil fauna such as  abundance in the case of soil samples/gradient extraction technique, activity/activity-densities in the case of pitfall traps/trap samples, also term density is often used in context of populations. Can authors clarify and specify it in „Methods section?

Line 81. Beta or alpha diversity of soil Collembola communities? Also inconsistent in the text of MS. Please, clearly specify second aim of your study.

Materials

For this subsection I suggest title „Study locality, plots and experimental design“

Line 87. (Fig. 1a,b) instead of (Fig. 1)

Line 88. experimental locality instead of experimental site

Line 88-97. I especially like the very accurate background informations on the research locality/plots, sampling design and environmental measurements, there is important to note more details for readers in section „Methods“. The description of methods section needs to be expanded. Please, give reference for regional climatic data of study area (temperature and precipitation). It is unclear which variables relate to air and which relate to soil (e.g., temperature can be related to both) in the text of ms. This should be clearly specified. „The average temperature -2.74 ℃“... own temperature measurements, continually measured soil or air temperature in 2018, or literary source? How was recorded vegetation structure in study locality (specialist, literary source...)?

Line 115. Study plots instead of study sites.

For this subsection I suggest title „Sampling and environmental viariables“

Lines 126-127.  Four sampling months/periods is more precisely than four sampling seasons, but I leave it on authors decision. Is Zhang et al. 2018 correct reference for this part?

Line 128. extracted individuals instead of organisms

Line 130 and 137. How was measured air temperature by thermometer/data-loggers in the OTCs and control plots from April to September? Continually? Are there single measurements for soil humidity and others edaphic factors? Please, specify it more precisely.

I suggest soil organic carbon (SOC) instead of SOC.

Data analyses

Line 144. activity-density vs. abundance. The use of term abundance for Collembola communities in the text of ms is more appropriate than activity-density, please, see also my comment above.

Line 156. Collembola communities instead of Collembola community.

Results

Why are basic data such as soil humid and soil-chemical data (SOC, NT, C/N) for individual plots/treatments not recorded in “Results”. Authors use them in analyses and in “Discussion” (Lines 229-231) they outline their importance, explain the higher spring community abundances of Collembola in terms of edaphic factors.  Please, I advise to add these useful data for readers.

For this subsection I suggest title „Collembola abundance and diversity“

Line 161. We identified 33 species of Collembola belonging to 15 genera across the different climate treatments. Please, hear you can add (Appendix B).

Lines 162-166. Please, delete redundant species authority for each species in this part, they are recorded in Appendix B.

Line 184. Please, remove here redundant Fig. 2b.

Line 192. Diversity instead of beta diversity.

Line 206- 210. How many species were included in RDA analysis and what criterion was used to select species for this analysis? Please, give this information. Soil organic carbon instead of soil organic matter and also total nitrogen instead of NT.   

Line 209. Please, remove redundantin the Tuqiang wetland in Northeast China“

Lines 218-219. Please, reformulate this sentence in Figure 4 more precisely, e.g. for the full names of the Collembola taxa see Appendix B.

Appendix B: Please, add code/abbreviations of Collembola species in Appendix B

Lines 212- 213. “Lepidocyrtus felipei was significantly affected by soil organic matter and root and shoot biomass” ...by root biomass, is it correct? Similarly, it is unclear for me that “Ceratophysella ainu was positively correlated with humidity “

Discusion

For first subsection I suggest title “The effect of warming on Collembola abundance and diversity”. Please, include also critical discussion about effect of warming on Collembola diversity (literary data and results of our study).

Lines 226-231. There is unclear first part of paragraph for readers, please reformulate it (what Collembola community structure changes..., previous study focused on...etc.).

Line 241. Remove redundant (Table 1), similarly Fig. 2a nd Table 2 (Lines 250-251).

Line 252. the effect of warming instead of the effect of species warming

Conclusions

Lines 288. „..long-term research on underground biota is needed”... I would like to advise to add relevant and recent reference in this context, useful for both authors and readers:

Roos, R. E., Birkemoe, T., Asplund, J., Ľuptáčik, P., Raschmanová, N., Alatalo, J., Olsen, S. L., Klanderud, K., 2020: Legacy effects of experimental environmental change on soil micro-arthropod communities. Ecosphere, 11(2): e03030, 1-17.

Appendix A. Please, add unit for Temperature (axis Y). More precisely, there is average temperature. “The average temperature of different treatments at different sampling times”

Author Response

Insects-response to “3rd response to editor comments”

Comment (abbreviated as C) 1: Dear authors, You have the potential of a good study here but have some work to do before this manuscript could be considered for publication. I hope the comments below help improve the study.

Response (abbreviated as R) 1: Thank you for the positive feedback and constructive suggestions to our study.

Second Response to the last Response to academic editor (abbreviated as RR)

SRR1: Done.

C2: It is a pity across the 6 years that these plots have been set up that no information of Collembola and the other variables (such as temp and humidity and plant growth etc) have been recorded. Showing how the treatment plots have changed compared (environment, plant assemblage etc) to the controls is a first step before looking at any changes in Collembola community.

R2: Done. Thanks a lot for your suggestions. You are right. Investigating the changes in the plant community and environmental factors during the first 1-3 years of warming is very important to interpret the response of the Collembola. Likewise, we believe that it is equally important to observe changes in plant communities, environmental factors and Collembola after long-term warming. Because in wetland ecosystems, the effects of warming on vegetation, soil properties and Collembola are in a state of stress fluctuation in the initial 1-3 years. After six years of warming, when environmental factors and vegetation communities stabilized, observing the stable effects of this long-term warming on plant communities, soil properties, and soil fauna can better reflect the final impact intensity of climate change. The wetland system is relatively fragile, and the early sample collection will have an important impact on the wetland vegetation and the Collembola. Therefore, in wetland ecosystems, the initial changes and long-term effects of warming on Collembola cannot be accurately investigated simultaneously in one experiment, and this study chose to focus on the latter.

SRR2: Done. We think the response is clear, no updated revisions.

C3: Specific comments: The title is misleading and should be revised to reflect what the study can reveal. At present the data supports an increase in surface activity of Collembola. Abundance and species richness. I agree with reviewer 1 that changes in abundance does not translate into a change in community structure.

R3: Done. We rephrased the sentence, replaced “benifits” with “increase the activity of”. Please see line 2.

SRR3: Done. We have revised the text as academic editor suggested in the last version, no updated revisions.

C4: Also, the authors do not specify what measure of “Species richness” they have used, currently it appears that the measure in Fig 2 and Table 1 is merely the actual number of species identified from the plots, which is not “species richness”. Other comments from reviewers also raised concerns regarding species identifications.

R4: Thank you for pointing this out. We have, accordingly, replaced “species richness” by “species richness (the number of species per sample)” in Fig 2 and Table 1.

SRR4: Done. We have revised the text as academic editor suggested in the last version, no updated revisions.

C5: I also have concerns given that only 9 species are identified of the 33 recorded, with 21 identified to genus and 1 only to Family. Of concern is how accurately these unidentified species recorded in different plots and also across different time periods of the study can be attributed to the same species.

R5: Thank you for pointing this out. Done. We have updated most of the species’ name in our study. The professional Collembola taxonomist Xin Sun, one of the co-author in this paper, had rechecked the specimens and sorted them to species level during these days. Please see Appendix B.

SRR5: Done. We have revised the text as academic editor suggested in the last version, no updated revisions.

C6: The authors use “growing season” rather than what has been suggested by the reviewer as the typical summer/autumn/winter/spring. While reading the text I found myself having to go back to work out whether the date coincided with warmer or cooler temperatures. Whether you use the standard ‘seasons’ or some other term it needs to be clearer for the reader.

R6: Done. We uniformly use “summer/autumn/winter/spring’” throughout the manuscript. Please see line 91-95.

SRR6: Done. We have revised the text as academic editor suggested in the last version, no updated revisions.

C7: Abstract Given the simulated warming was undertaken across 6 years (5 years plus the 1 year of collecting Collembola in the current study) the authors missed a tremendous opportunity to have undertaken a multi-year study here that would have been of much greater significance.

R7: Thank you for your suggestions. The wetland system is relatively fragile, and the early sample collection will have an important impact on the wetland vegetation and the Collembola. Therefore, in wetland ecosystems, the initial changes and long-term effects of warming on Collembola cannot be accurately investigated simultaneously in one experiment, and this study chose to focus on the latter. We will increase the replications to conduct the multi-years’ investigation in the future study.

SRR7: Done. We have revised the text as academic editor suggested in the last version, no updated revisions.

C8: Result 1, I do not see that your results support a change in “community structure” in different growing season”.

R8: We have clarified the first result as follows (line 165-167): “warming treatment affected the species richness and abundance of Collembola in the different growing season”

SRR8: Done. New methods were adopted to analyse the effect of warming on Collembola community structure. Please see line 188-200 and fig. 4.

C9: For the 3rd result you did not measure changes in food resources across seasons so your comment that it had no apparent impact in this region is complete speculation. I would assume warming and greater water availability would have an impact on food resources, especially if you take into account greater activity (which might require greater food availability). I am also concerned about interpretations about the Collembola community when only 9 of 33 species can be identified to species-level.

R9: Done. We have revised the sentence in the text. Please see line 278-280.

For the species identification, we have updated most of the species’ name in our study. The professional Collembola taxonomist Xin Sun, one of the co-author in this paper, had rechecked the specimens and sorted them to species level during these days.

SRR9: Done. We have revised the text as academic editor suggested in the last version, no updated revisions.

C10: Your 2nd result is the main outcome from the study, showing that activity is increased with temperature. However, you can not interpret “activity” as an increase in Collembola density in the plots.

R10: Done. Thanks for your suggestions. We have changed the word “density” to “activity-density”. We think Collembola captured in the same area in all the replicates by aspirator do reflect the aspect of density (especially for ground dwelling Collembola). Therefore, the word “activity-density” was used in the text. Please see line 159, 165,168-169.

SRR10: Done. We have revised the text as reviewer 2 suggested, we relace “density” to “abundance”. Please see document “4th response to reviewer” C37 and R37.

C11: General text As the reviewers have already identified the text requires a thorough reading and editing for grammar and spelling.

R11: Done, grammar and spelling were carefully checked and revised. please see the certification and text.

SRR11: Done. We have revised the text as academic editor suggested in the last version, no updated revisions.

C12: Some things I picked up but this is not an exhaustive list: Line 44: is “Northeast China” an official region? If not it should be ‘northeast’ here and elsewhere in ms. Note it is “northeastern” at line 77.

R12: Done, “Northeast China” is an official region. We revised them. please see line 172, 206, 214, 217, 224, 293.

SRR12: Done. We have revised the text as academic editor suggested in the last version, no updated revisions.

C13: Line 44: “Many studies have simulated climate change but mainly…”

R13: Done, please see line 45.

SRR13: Done. We have revised the text as academic editor suggested in the last version, no updated revisions.

C14: Line 47: “…wetlands have received less attention”

R14: Done, please see line 48.

SRR14: Done. We have revised the text as academic editor suggested in the last version, no updated revisions.

C15: Line 51-52: what is a “free-running life cycle”??

R15: Done, we revised the text. please see line 52.

SRR15: Done. We have revised the text as academic editor suggested in the last version, no updated revisions.

C16: Line 53: “…as most collembolan species display all age (instar) stages throughout…” note lowercase “collembolan”

R16: Thank you for your suggestions. The word “collembolan” is right.

SRR16: Done. We have revised the text as academic editor suggested in the last version, no updated revisions.

C17: Line 55: do you mean “high-latitude Arctic…”??

R17: Yes. Thank you for your suggestions. Done, please see line 55.

SRR17: Done. We have revised the text as academic editor suggested in the last version, no updated revisions.

C18: Line 65: “However, global…”

R18: Done, please see line 66.

SRR18: Done. We have revised the text as academic editor suggested in the last version, no updated revisions.

C19: Line 72: “warming could further affect…” I assume changes have already occurred given the statement at line 66?

R19: Thank you for your suggestions. Done, please see line 73.

SRR3: Done. We have revised the text as academic editor suggested in the last version, no updated revisions.

C20: Lines 76-78: examining the “response” to warming is actually since 2013, you collect the data in 2018.

R20: Thank you for your suggestions. Done, please see line 77-78.

SRR21: Done. We have revised the text as academic editor suggested in the last version, no updated revisions.

C21: Line 90: “Sphagnum spp.,”

R21: Thank you for your suggestions. Done, please see line 97.

SRR21: Done. We have revised the text as academic editor suggested in the last version, no updated revisions.

C22: Line 94: “…60-80 cm below ground level,…”

R22: Thank you for your suggestions. Done, please see line 101.

SRR22: Done. We have revised the text as academic editor suggested in the last version, no updated revisions.

C23: Line 95: “permafrost layer ranged 80-150 cm above the surface.” I don’t understand this, above the surface would be in the air?? Or do you mean above the transition layer??

R23: Thank you for your suggestions. Done, please see line 101-102.

SRR23: Done. We have revised the text as academic editor suggested in the last version, no updated revisions.

C24: Line 96: “…where soil samples…”

R24: Thank you for your suggestions. Done, please see line 103.

SRR24: Done. We have revised the text as academic editor suggested in the last version, no updated revisions.

C25: Line 101: Reference 29 cited here is not correct.

R25: Done, reference was cited in wrong place. We delete the sentence and reference.

SRR25: Done. We have revised the text as academic editor suggested in the last version, no updated revisions.

C26: Line 108: incorrect reference cited (it should not be 30).

R26: Thank you for your suggestions. It is a fault. We delete the sentence and reference.

SRR26: Done. We have revised the text as academic editor suggested in the last version, no updated revisions.

C27: Line 110: Surface Collembola will be a mix of surface (mostly) and some soil surface Collembola. You can not refer to them as soil dwelling Collembola. Given the introduction focuses only on “soil” some revision of the introduction is required because you don’t actually measure any species from the soil. Currently no information has been provided on the vegetation differences between each treatment plot and also to each control plot, just a general vegetation for the region.

R27: Thank you for your suggestions. We have changed “surface soil dwelling” to “ground dwelling”. Please see line 119.  We change some reference to focus on the related function of Collembola in wetlands. Please see line 51-52. Furthermore, wetlands are seasonal waterlogged. Hemiedaphic Collembola still perform certain ecological functions in soil in dry seasons. Some sentences were left to describe the relationship between Collembola and soil.  Plant community did not change under warming conditions. That is why we describe the general vegetation in our study region.  

SRR27: Done. We have revised the text as academic editor suggested in the last version, no updated revisions. Reviewer 1 also commented the sampling method of Collembola. Please see document “4th response to reviewer” C4 and R4 “Aspirator was used in our experiment. The front end of the aspirator is a square iron frame  (modified air blower, Stroke 4-GX 35, Honda, sampling diameter of opening = 14 cm). The staying time of the aspirator has been investigated in the former studies, which can ensure the absorption of the springtails below the iron frame area. Our study area located in a wetland with a certain degree of water accumulation on the surface, mainly composed of ground-dwelling springtails with strong mobility. The soil drills collected mainly euedaphic groups, so we chose to use the aspirator to collect the Collembola groups that appeared in this experiment. For details, please see attachment”.

C28: Line 113: Why measure temperature at 50 cm above the ground? That is not where the Collembola live, surface temperature if you have it would be more suitable for this study.

R28: Thank you for your suggestions. Wetlands were waterlogged seasonally. On the one hand, temperature measuring instruments are easily submerged and damaged by water; on the other hand, water can buffer the effect of warming, which makes the results inaccurate. The effect of warming is mainly reflected in the indirect effect on Collembola through plants variation. Therefore, placing it at 50 cm is mainly to reflect the effect of warming on plants, thereby reflecting the changes in Collembola.

SRR28: Done. We have revised the text as academic editor suggested in the last version, no updated revisions. Reviewer 1 also commented the sampling method of Collembola. Please see document “4th response to reviewer” C7 and R7 “The research field of this experiment is a wetland, with a certain amount of accumulated water on the surface, and little "soil", which is basically composed of plant roots. The effect of warming on the temperature of "soil" should be the effect of warming on water. This could buffer the effect of warming. Therefore, we tested the temperature that has a greater impact on wetland plants to reflect the effect of warming. Moreover, the effect of warming on ground-dwelling springtails were mainly caused by the change of plant properties in wetlands”.

C29: Lines 114-115: Why have you not measured temperature for the whole study period? And if measured from May to September why only give the average difference from June-Sept? I am surprised temperature has not been recorded in the 12 plots for the 6 years since they were established.

R29: Done. Thanks for your suggestion. This is very helpful for us treating with experiments in the future. We actually have another experiment in the same study area. It was an OTC warming experiment that focused on the effects of warming on plants, microbes and greenhouse gas emissions (Cui, Q., Song, C., Wang, X., Shi, F., Yu, X., Tan, W., 2018. Effects of warming on N2O fluxes in a boreal peatland of Permafrost region, Northeast China. Science of the Total Environment 616-617, 427-434). In the short-term experiment, the temperature in warming and control treatment was monitored from the beginning. We found that the results were basically consistent with the monitoring results of our experiment in 2018. Therefore, we used the monitoring results of 2018. But it is true that we did not monitor the data of the 5 years in this long-term experiment. While in 2018, all the 12 OTC plots were monitored, we have supplemented Appendix A to describe the results for temperature.

SRR29: Done. We have revised the text as academic editor suggested in the last version, no updated revisions.

C30: Lines 117-121: this information doesn’t belong in “Sampling and analysis”. Also, remove the date and use the month, otherwise revise the English here. Why provide the avg temp for April, but for no other season? I would personally use “germinate” rather than “sprout”.

R30: Done, we put them in the part of “Study sites and experimental design”. We replace “sprout” with “germinate”. Please see line 91-96.

SRR30: Done. We have revised the text as academic editor suggested in the last version, no updated revisions.

C31: Line 121: Clearly there are not 4 growing seasons?

R31: Done, please see line 126.

SRR31: Done. We have revised the text as academic editor suggested in the last version, no updated revisions.

C32: Lines 121-123: it is a pity the experimental design did not include Collembola being collected when the plots were set up in 2013 (as a baseline) or any years between then and the collections in 2018.

R32: Thanks for your suggestions. We will consider it in the future experiment.

SRR32: No updated revisions.

C32: Line 123: does the aspirator really have an opening 14 cm wide? That is very wide!

R32: Yes. The aspirator has connected to an external 14cm steel frame. The large diameter could match the strong moving abilities of ground dwelling Collembola.

SRR32: Done. We have revised the text as academic editor suggested in the last version, no updated revisions. Reviewer 1 also commented the sampling method of Collembola. Please see document “4th response to reviewer” C4 and R4 “Aspirator was used in our experiment. The front end of the aspirator is a square iron frame (modified air blower, Stroke 4-GX 35, Honda, sampling diameter of opening = 14 cm). The staying time of the aspirator has been investigated in the former studies, which can ensure the absorption of the springtails below the iron frame area. Our study area located in a wetland with a certain degree of water accumulation on the surface, mainly composed of ground-dwelling springtails with strong mobility. The soil drills collected mainly euedaphic groups, so we chose to use the aspirator to collect the Collembola groups that appeared in this experiment. For details, please see attachment”.

C33: Line 125: Both the references “Christiansen and Bellinger [35], and Hopkin [36]” are incorrect, in the reference list these are 29 and 30. ALL your references need careful checking!

R33: Thank you for your suggestions. We rechecked all the reference, please see line 322-429.

SRR33: Done. We have revised the text as academic editor suggested in the last version, no updated revisions.

C34: Line 129-130: The use of “plant growth and plant withered” and mixing between “season” and “period” is cumbersome. This needs revising to a more standard and easier to follow system such as summer/winter etc.

R34: Thank you for your suggestions. Done, we revised as you suggested. Please see line 91-96.

SRR34: Done. We have revised the text as academic editor suggested in the last version, no updated revisions.

C35: Lines 129-133: I can’t follow here, did you cut all the plants from the plots in July and Sept? won’t this adversely affect the Collembola?

R35: Thank you for your suggestions. The entire plot area is 100cm*100cm, our sampling area is 10cm*10cm. We revised them in the text. Please see line 134.

SRR35: Done. We have revised the text as academic editor suggested in the last version, no updated revisions.

C36: At Lines 113-114 the plant growing season is May-Sept, but here you are saying that September is “plant withered period”?

R36: Thank you for your suggestions. We have changed the plant growth period to season as previously suggested. Please see line 91-96.

SRR36: Done. We have revised the text as academic editor suggested in the last version, no updated revisions.

C37: Line 135: Species abundance and richness dominate the results, but commonly used measure for such studies has been Shannon's diversity index, this should be included (or an alternative ‘diversity’ index).

R38: Thank you for your suggestions. We supplement the results of the Shannon Wiener Index, please see table 1. The results of the diversity index are the same as the results of the number of species, so we did not show this part of the results in an independent figure.

SRR38: Done. We have revised the text as academic editor suggested in the last version, no updated revisions.

C39: Lines 154-156: At first mention of species the authorities need to be included. Also, you have 2 species of Sminthurides in Table A, but just mention Sminthurides sp. here, do you mean both species (then it should be spp.) or something else? There is very little information here, what were the abundant species in the controls? More descriptive analysis on how the species differ across the seasons and between the control and treatment plots is lacking.

R39: Thank you for your suggestions. Done, we have revised the species name as you suggested. The distribution differences of Collembola between warming and control treatments were also provided in the text, please see line 160-164.

SRR39: Done. We have revised the text as academic editor suggested in the last version, no updated revisions.

C40: Line 163: “'*' indicates significance at p ≤ 0.05;”; not needed in the caption as it doesn’t appear in the Table. Here and elsewhere “p” should be in italics (it is a statistic).

R40: Done. We have revised the text as you suggested. Please see line 173-175.

SRR40: Done. We have revised the text as academic editor suggested in the last version, no updated revisions.

C41: Lines 165-172: The results cited here are not all presented in Table 1 and are only presented as a graphic in Fig 2. For example, the statement “In the control treatments in May, species richness was significantly higher than that in April, September, and August (Fig. 2b).” is not supported by the figure because the reader can not determine if the plot shown is “significant” or not.

R41: Thank you for your suggestions. We provided a sentence to describe the significant difference in the figure. Please see line 186.

SRR41: Done. We have revised the text as academic editor suggested in the last version, no updated revisions.

C42: Figure 2: I can not find any explanation on what the capital letters are representing at the top of the plots in Fig 2a and 2b. I also would be interested in knowing if the, on average, 2 or 3 treatment plots that are greater than the controls if the species composition varies? Because in 3 or 4 plots for each of the 4 sampling periods the treatment plots are not different to the controls, this should be explored further. I was expecting to find data here for the temperature and humidity for each plot across the 4 seasons. The only information on this is the statement at lines 114-115 that the treatment plots were on average 1.23 C higher than the controls but only for June to Sept. For an artificially induced warming experiment you need to first establish the pattern of the warming (temp and humidity) for each treatment and control plot (individually) for each sampling period, this has not been done.

R42: Done. We have clarified the capital letters in the legend of Fig 2 “Bars with the same letter do not differ significantly (p > .05; Tukey's HSD test).”

We have clarified the variation of species composition in the Figure 3.

We have included the temperature in our study, please see Appendix A. In wetland, it is often waterlogged. Therefore, it was measured only in July for redundancy analysis (RDA).

SRR33: Done. We have revised the text as academic editor suggested in the last version, no updated revisions. Furthermore, we have supplemented “Table 1” to show the environmental and plant properties in OTC and control treatment. Please see line 155-157.

C43: Line 178 and 185-187: this is the first mention of a “PERMANOVA”, needs to be in methods also, and where exactly are these results? Figure 3: This figure needs to be in higher resolution. I also do not see what the authors claim in lines 179-185, there might be some separation for the May samples, but in all the others the OTC samples are merely a subset of the controls, and while one axis may explain more of the variation than the other for some of the treatment plots the authors do not expand on the significance of this. How much variation was explained by each axis? And did the authors explore the 3rd axis or exclude because most of the variation was explained by the first 2?

R43: Done. We now included the details about PERMANOVA in the method and exactly result. Lines 148-150: “A PERMANOVA was used to quantify differences in collembolan community composition between warming and control treatments using the adonis function within the vegan package.” Lines 188-189: “Permutational multivariate analysis of variance (PERMANOVA) showed that warming significantly affected the community composition of Collembola (F = 1, P = 0.004).”

We have increased the resolution of Figure 3.

We give the significantly PERMANOVA result of community composition between control and warming treatment, lines 195-197: “PERMANOVA confirmed the significant difference in species composition between warming and control in Spring (R2 = 0.363, P = 0.003) and Autumn (R2 = 0.259, P = 0.004).”

NMDS, as an iterative algorithm, choosing the appropriate number of dimensions is an interactive process. After the initial ordination, we examined the stress values generated by the algorithm. As a rule of thumb, an NMDS ordination with a stress value around 0.1 is considered fair. In our study, using two axes can visualize the community data well.

SRR43: Done. Data were reanalysed and species name were included in fig. 4. We also rephrased the results give more information about species. Please see line 247-273.

C44: Fig 4. I like this plot and great to see Table 2. The link between temperature and humidity makes sense. However, this plot/data by itself is not very informative. The data was collected on 2 occasions (April and Sept) and these should be shown separately to gauge the difference. Also, Table 2 combines all OTC and control plots together, this makes no sense at all. The control and treatment plots should be dealt with separately and RDA plots shown separately also (which are missing entirely for the control plots).

R44: Thank you for your suggestions. Temperature data were monitored at all sampling times, but SOC, nitrogen, moisture, root and shoot biomass was only measured once in July. Our RDA analysis utilized environmental and plant data once to encompass the effects of multiple factors on Collembola communities. Therefore, plotting control and warming separately does not present well (see figures as follows). Taking your comments into account, we added control and warming as nominal variables to the overall RDA analysis for the treatment factors, which allows us to understand the factors, such as humidity, SOC and total nitrogen, are altered by warming to affect Collembola community structure.

Figures. RDA in warming and control plots

SRR44: Done. We have redone the RDA as reviewer suggested in the latest version (see document “4th response to reviewer” C3 and R3, C19 and R19). Please see Figure 5 and table 3. Data were reanalysed and species name were included in fig. 4. We also rephrased the results give more information about species. Please see line 247-273.

C45: Lines 213-216: This description is needed in the methods, and highlights that the authors are looking at surface litter Collembola and not soil Collembola. This is also indicated by the genera identified in this study. Line 223: “collembolan”

R45: Thank you for your suggestions. we add the descriptions in the methods please see line 113-120. The word “collembolan” can be used as an adjective or noun of “Collembola”. It is right here.

SRR45: Done. We have revised the text as academic editor suggested in the last version, no updated revisions.

C46: Line 224: I don’t understand the reference to food sources here helping eggs to hatch earlier?

R46: Thank you for your suggestions. we revise the sentence in the text, please see line 227-230.

SRR46: Done. We have revised the text as academic editor suggested in the last version, no updated revisions.

C47: Lines 228-229: not just warming, humidity also, clearly these will be linked variables but in the treatment plots humidity is increased by the lower wind also.

R47: Thank you for your suggestions. We revised as you suggested, please see line 232-234.

SRR47: Done. We have revised the text as academic editor suggested in the last version, no updated revisions.

C48: Lines 231-242: Without providing data on the control plots and separately for the 2 collections made it is difficult to make any comparisons here. Line 264-272:

R48: Thank you for your suggestions. We revise the figure and the text, please see figure 4 and line 202-211.

SRR48: Done. We have revised the text as academic editor suggested in the last version, no updated revisions.

C49: Discussion on soil moisture, although important, but you did not measure soil moisture in your study.

R49: Thank you for your suggestions. Soil moisture was also monitored once in July for RDA.

SRR49: Done. we have supplemented “Table 1” to show the environmental and plant properties in OTC and control treatment. Please see line 155-157.

Reviewer 4 Report (New Reviewer)

The manuscript “Collembola at three alpine subarctic sites resistant to twenty years of experimental warming” deals with a very interesting phenomenon. The overall study is relatively straightforward and the data and conclusions are also valuable. The experiment is well set up, and I appreciate the supplementary material video presenting the sampling procedure. The effort of the authors to describe the results to the community and individual species level is also commendable.

I would have liked to read some examples in the discussion about possible interactions between certain species. The change in the dominance structure is interesting, some species benefit from the changed conditions, while others may disappear or respond with a decrease in numbers due to the increased dominance of certain euryok species. If the authors mentioned some such examples (from the own results or literature data), it could slightly improve the manuscript.

Author Response

Reviewer 4

The manuscript “Collembola at three alpine subarctic sites resistant to twenty years of experimental warming” deals with a very interesting phenomenon. The overall study is relatively straightforward and the data and conclusions are also valuable. The experiment is well set up, and I appreciate the supplementary material video presenting the sampling procedure. The effort of the authors to describe the results to the community and individual species level is also commendable.

Comment 23: I would have liked to read some examples in the discussion about possible interactions between certain species. The change in the dominance structure is interesting, some species benefit from the changed conditions, while others may disappear or respond with a decrease in numbers due to the increased dominance of certain euryok species. If the authors mentioned some such examples (from the own results or literature data), it could slightly improve the manuscript.

Response 23: Done. We have included more details about the changes in species composition in the discussion and discussed the mechanisms and why it applies to only this species and not the others, lines 237-258.

Round 2

Reviewer 1 Report (New Reviewer)

The manuscript contains a lot more detail that was missing in the previous version and that is great! However, the overall discussion and arguments for why certain choices were made is still below the level of what I would expect in a scientific article. 

Abstract:

warming treatment 29 affected the species richness and abundance of Collembola in the different growing season” how in what  way/manner?

increase Collembola abundance” by how much?

the Collembola community composition in perma-32 frost wetlands is mainly determined by air humidity” This is a really unexpected finding given the overall start of the abstract. In addition, how was the community affected?

Introduction; all the relevant information is here but there is a general lack of cohesion in the paragraphs. Each paragraph should have a coherent message. E.g, the para starting on line 49 starts with the effect of collembola on ecosystem processes but ends with their response to climate warming manipulations. There is no logical connection between the two and considering that this particular study only deals with the collembola response there is no need to make one. However, that does leave the reader with a very unusual line of arguments in the introduction that do not logically lead to the hypotheses. For example, nowhere in the introduction is there any clue as to why the collembola community would change under warming and by what mechanism.

Unclear what is means by the following sentence: “Only several studies concentrated the response of soil fauna to climate 47 change in middle and high latitude wetlands [4-6].” There are indeed several studies and far more than those referred to by 4, 5 and 6. What makes this specific study of interests? Is that the location or are you focusing on a specific subset of animals?

Lines 49-51 would be stronger if quantitative effects are mentioned. I agree in general sense but the argument would be stronger with statements that include quantitative effects.

CANOCO Version 4.5 (ter Braak and Šmilauer, 2002) was used for data analysis. A 148 multivariate redundancy analysis (RDA) was used to analyze the effect of environmental 149 parameters including temperature and humidity, and root biomass, shoot biomass and 150 TN on soil Collembola community.soil organic carbon and total nitrogen on soil Collem-151 bola communities” How does this work with the repeated measures on collembola while plant and other variables were measured only once? These variables were not consistent through time so it is unfair/incorrect to keep these stable while the collembola data changes over time.

When describing significant differences in results text I think it is best to include actual data or provide % differences. That way readers get a better insight in the quantitative changes that are affected. For instance, your root biomass doubles, and this is a considerable change and has large ecological consequences. Air temperature however, rises from 18.6 to 19.8 which is about 6% increase, and therefore, not equivalent to the change in root biomass.

Table 1 please indicate number of replicates for each treatment. What do the different letters mean next to each value? Take care with decimals, this needs to reflect the accuracy of the measurements; did your temperature loggers have an accuracy down to 3 decimals? (I very much doubt this)

I don’t understand the following sentence: “After warming treatment, Collembola abundance increased sig-170 nificantly in April and September (Fig. 3).” What does after warming treatment mean?

Table 2 no need to include ‘*’ if actual p-values are shown.

Figure 2 would suggest you group bars (not box plots) per season; so control and OTC of April are next to one another (smaller spacing compared to difference between April and May). So that differences between season and treatments are clearer. At present all appear equal but that is not correct.  

Figure 3 seems superfluous considering that the same can be derived at from figure 2 if the bars are better grouped.

Use of ‘after warming treatment’ is really confusing as the warming did not actually stop. OTCs were present continuously so either state that response variables differed, in a quantitative manner, between control and OTC or in response to warming but not ‘after warming’.

The light yellow/orange coloring is difficult to read so please change for clearer contrast.

In the para on community response the following is often used ‘were dominant after warming treatments’ but what does this mean? Do the species actually have a higher abundance or do the stand out  because of different relative abundance? This has different ecological meaning/consequences.

Changes in temperature and humidity caused by warming conditions affected the 208 community structure of Collembola.” None of the above actually supports this. So be careful with the wording here. At most you could state that there was an association between the observed changes in the collembola community composition and the changes in air humidity and temperature. Considering table 3 and the accompanying results text it is not even correct or at least only a small part of the story.

Discussion section 4.1 indicates that this deals with “The effect of warming on abundance and species richness of Collembola” However, all focus lies on the variation through time. This para should be focused on the observed pattern in collembola abundance and richness in comparison to the treatment effects in the OTCs and how this differs between other warming studies.

Discussoin 4.2 is very much a summary of the results section without a proper discussion of these findings in relation to existing literature. What do you want your readers to take away from this study in light of the question that is addressed here?

ing treatment, the community composition of Collembola became more aggregated (the 254 circle of OTC became smaller), which might be due to some species disappearing because” surely this information is available to the authors?

Spe-261 cifically, Sminthurinus aureus more abundant in the control treatment, the species might 262 be sensitive to the warming [39]” Why would this species be more sensitive? What is the mechanisms and why does it apply to only this species and not the other ones?

Some experimental warming studies conducted at high latitudes showed that the re-279 sponse of Collembola to warming is very different” different in what way?

You did not quantify soil humidity so cannot conclude that soil  moisture was affected in a positive or negative manner for the collembola. In addition, it is odd that conditions measured at 50 cm above the soil are relevant for species that live in the soils and among the litter layer.

The findings of this study indicated that warming increases the Collembola abun-300 dance in permafrost wetlands” but we still do not know by how much and it doesn’t become clear whether this changes is different compared to other systems and with respect to the changes in the environmental variables.

Author Response

Reviewer 1

Comment 1: The manuscript contains a lot more detail that was missing in the previous version and that is great! However, the overall discussion and arguments for why certain choices were made is still below the level of what I would expect in a scientific article.

Response 1: Thank you so much for your patience and guidance. We are really appreciated for your valuable comments and we learned a lot during the progress. We have improved the overall discussion according to the new results. Lines 217-279.

Comment 2: Abstract: “warming treatment affected the species richness and abundance of Collembola in the different growing season” how in what way/manner?

Response 2: Done. We have specified that “warming treatment increases the species richness and abundance of Collembola in the different seasons, except in May.”

Comment 3: Line 31: “increase Collembola abundance” by how much?

Response 3: Done. We included some details in April, the abundance increased about eight times.

Comment 4: Line 32: “the Collembola community composition in permafrost wetlands is mainly determined by air humidity” This is a really unexpected finding given the overall start of the abstract. In addition, how was the community affected?

Response 4: Done. We included the changes in species composition in different seasons now, lines 185-194. Moreover, we had the response of different species to air humidity in discussion, lines 237-242.

Comment 5: all the relevant information is here but there is a general lack of cohesion in the paragraphs. Each paragraph should have a coherent message. E.g, the para starting on line 49 starts with the effect of collembola on ecosystem processes but ends with their response to climate warming manipulations. There is no logical connection between the two and considering that this particular study only deals with the collembola response there is no need to make one. However, that does leave the reader with a very unusual line of arguments in the introduction that do not logically lead to the hypotheses. For example, nowhere in the introduction is there any clue as to why the collembola community would change under warming and by what mechanism.

Response 5: Done. We revised manuscript as reviewer suggested. We rearranged the order of the sections, taking into account the connectivity of each section, lines 46-53; 64-69. The mechanism of how warming affects soil fauna such as springtail community were also added in the text, lines 51-53.

Comment 6: Introduction, Line 47: Unclear what is means by the following sentence: “Only several studies concentrated the response of soil fauna to climate change in middle and high latitude wetlands [4-6].” There are indeed several studies and far more than those referred to by 4, 5 and 6. What makes this specific study of interests? Is that the location or are you focusing on a specific subset of animals?

Response 6: Done. We revised manuscript as reviewer suggested, lines 51-53.

Comment 7: Introduction, Lines 49-51 would be stronger if quantitative effects are mentioned. I agree in general sense but the argument would be stronger with statements that include quantitative effects.

Response 6: Done. We revised sentence as reviewer suggested, lines 54-55.

Comment 7: Method: Lines 147-152: “CANOCO Version 4.5 (ter Braak and Šmilauer, 2002) was used for data analysis. A multivariate redundancy analysis (RDA) was used to analyze the effect of environmental parameters including temperature and humidity, and root biomass, shoot biomass and TN on soil Collembola community. soil organic carbon and total nitrogen on soil Collembola communities” How does this work with the repeated measures on collembola while plant and other variables were measured only once? These variables were not consistent through time so it is unfair/incorrect to keep these stable while the collembola data changes over time.

Response 7: Thanks a lot for your comment. We have to confess that it could be better to match the environmental factors and Collembola community in each sampling time. However, our study was conducted in a natural peatland with a warming treatment. Due to the specificity of the experiment, we can not repeat the assessment of root biomass in each season; otherwise, the plants in the OTC could be destroyed totally and Collembola could be impacted severely. We put the number of collembola in all growth periods together, because the change of the whole community is the result of accumulation in different periods. The plant biomass and environmental factors investigated in the last time are also the result of the final accumulation of the whole growth period. The accumulation value of collembola abundance and accumulated values of environmental factors completed were used for the RDA.

Comment 8: Result 3.1: When describing significant differences in results text I think it is best to include actual data or provide % differences. That way readers get a better insight in the quantitative changes that are affected. For instance, your root biomass doubles, and this is a considerable change and has large ecological consequences. Air temperature however, rises from 18.6 to 19.8 which is about 6% increase, and therefore, not equivalent to the change in root biomass.

Response 8: Done. We have included more details. Lines 153-156.

Comment 9: Result 3.1: Table 1 please indicate number of replicates for each treatment. What do the different letters mean next to each value? Take care with decimals, this needs to reflect the accuracy of the measurements; did your temperature loggers have an accuracy down to 3 decimals? (I very much doubt this)

Response 9: Done. “The different letters next to each value represent significant differences (p < 0.05)”. We double-checked the accuracy of the temperature loggers, and we kept the accuracy down to one decimal. Please see line 158 and table 1.

Comment 10: Result 3.2, line 170: I don’t understand the following sentence: “After warming treatment, Collembola abundance increased significantly in April and September (Fig. 3).” What does after warming treatment mean?

Response 10: Thank you for your comments, we uniformly use “in warming treatment” throughout the manuscript.

Comment 11: Table 2 no need to include ‘*’ if actual p-values are shown.

Response 11: Done, we have removed “*”.

Comment 12: Figure 2 would suggest you group bars (not box plots) per season; so control and OTC of April are next to one another (smaller spacing compared to difference between April and May). So that differences between season and treatments are clearer. At present all appear equal but that is not correct. Figure 3 seems superfluous considering that the same can be derived at from figure 2 if the bars are better grouped.

Response 12: Thank you for your suggestion. We have replotted figure 2 and removed Figure 3.

Comment 13: Use of ‘after warming treatment’ is really confusing as the warming did not actually stop. OTCs were present continuously so either state that response variables differed, in a quantitative manner, between control and OTC or in response to warming but not ‘after warming’.

Response 13: Thank you for your comments, we uniformly use “in warming treatment” throughout the manuscript.

Comment 14: The light yellow/orange coloring is difficult to read so please change for clearer contrast.

Response 14: We have changed the colour for OTC and CK treatment consistent with Figure 2.

Comment 15: Result 3.3: In the para on community response the following is often used ‘were dominant after warming treatments’ but what does this mean? Do the species actually have a higher abundance or do the stand out because of different relative abundance? This has different ecological meaning/consequences.

Response 15: We have modified to “were dominant in warming treatments (with higher abundance)”, it’s means with higher abundance not relative abundance.

Comment 16: Result 3.4: “Changes in temperature and humidity caused by warming conditions affected the community structure of Collembola.” None of the above actually supports this. So be careful with the wording here. At most you could state that there was an association between the observed changes in the collembola community composition and the changes in air humidity and temperature. Considering table 3 and the accompanying results text it is not even correct or at least only a small part of the story.

Response 16: we delete this incorrect sentence.

Comment 17: Discussion section 4.1 indicates that this deals with “The effect of warming on abundance and species richness of Collembola” However, all focus lies on the variation through time. This para should be focused on the observed pattern in collembola abundance and richness in comparison to the treatment effects in the OTCs and how this differs between other warming studies.

Response 17: Thank you for your comments. We have revised as reviewer suggested, lines 221-232.

Comment 18: Discussion 4.2: “After the warming treatment, the community composition of Collembola became more aggregated (the circle of OTC became smaller), which might be due to some species disappearing because” surely this information is available to the authors?

Response 18: Thank you for your comments. We have modified this discussion. We would like to provide the authors that some species benefit from the changed conditions, while others may disappear or respond with a decrease in numbers due to the increased dominance of certain Collembola species.

Comment 19: Discussion 4.2 “Specifically, Sminthurinus aureus more abundant in the control treatment, the species might be sensitive to the warming [39]” Why would this species be more sensitive? What is the mechanisms and why does it apply to only this species and not the other ones?

Response 19: Thank for your comments, we have provided more explanation about the result, Sminthurinus aureus might be sensitive to the warming, this may be related to the growth and reproduction of the species.

Comment 20: Discussion 4.3“Some experimental warming studies conducted at high latitudes showed that the response of Collembola to warming is very different” different in what way?

Response 20: Done, we have reformulated this sentence, lines 265-266.

Comment 21: You did not quantify soil humidity so cannot conclude that soil moisture was affected in a positive or negative manner for the collembola. In addition, it is odd that conditions measured at 50 cm above the soil are relevant for species that live in the soils and among the litter layer.

Response 21: Done, we add the sentence which describe the RDA result related to humidity and abundant Collembola species, lines 275-279. Because putting sensors into soil (actually into water) in the wetland system will influence the warming effect on temperature. In order to unify the conditions of temperature and humidity, we placed the temperature and humidity sensors in the air 50cm above the ground to reflect changes in temperature and humidity simultaneously.

Comment 22: Conclusion: “The findings of this study indicated that warming increases the Collembola abundance in permafrost wetlands” but we still do not know by how much and it doesn’t become clear whether this changes is different compared to other systems and with respect to the changes in the environmental variables.

Response 22: Done. We included that warming increases the Collembola abundance in permafrost wetlands (about eight times in April). Additionally, we also discussed the effect of warming on permafrost wetlands and other systems.

This manuscript is a resubmission of an earlier submission. The following is a list of the peer review reports and author responses from that submission.

Round 1

Reviewer 1 Report

  • table 1, the legends must be identical to the content of the table (e.g. **0.01 in legends - 0.001 in the table body)
  • line 265 ..."Mesozoic fauna"... This must be a mistake, please check.

Author Response

Comment 1:

table 1, the legends must be identical to the content of the table (e.g. **0.01 in legends - 0.001 in the table body)

Response 1:

Done, we revised this manuscript as the reviewer suggested. Please see line (159)

Comment 2:

line 265 ..."Mesozoic fauna"... This must be a mistake, please check.

Response 2:

Done, we revised this manuscript as the reviewer suggested. Please see line (265)

Reviewer 2 Report

The authors made substantial modifications to the papaer, that can be now accepted

Author Response

Comment 1:

The authors made substantial modifications to the paper, that can be now accepted

Response 1:

Thank you very much for your nice comments on our manuscript.